# A country-wide health policy in Chile for deaf adults using cochlear implants: Analysis of health determinants and social impacts

**Mario Bustos-Rubilar**[1,2]*, **Fiona Kyle**[1], **Eliazar Luna**[3], **Kasim Allel**[4], **Ximena Hormazabal**[5], **Daniel Tapia-Mora**[2,6], **Merle Mahon**[1]

**1** Division of Psychology and Language Science, University College London, London, United Kingdom, **2** Departamento de Fonoaudiología, Facultad de Medicina, Universidad de Chile, Santiago, Chile, **3** Department of Clinical, Educational and Health Psychology, University College London, London, United Kingdom, **4** Institute for Global Health, University College London, London, United Kingdom, **5** Carrera de Fonoaudiología, Departamento de Ciencias de la Salud, Pontificia Universidad Católica de Chile, Santiago, Chile, **6** Escuela de Fonoaudiología, Universidad de los Andes, Santiago, Chile

* mario.rubilar.18@ucl.ac.uk

**Data Availability Statement:** All relevant data are within the paper and its Supporting Information files.

## Abstract

### Background

Post-lingual deafness represents a critical challenge for adults' well-being with substantial public health burdens. One treatment of choice has been cochlear implants (CI) for people with severe to profound hearing loss (HL). Since 2018, Chile has implemented a high-cost policy to cover CI treatment, the "Ley Ricarte Soto" (LRS) health policy. However, wide variability exists in the use of this device. To date, no related study has been published on policy evaluation in Chile or other Latin American countries.

### Objectives

This study aimed to evaluate the impact of the LRS policy on the treatment success and labour market inclusion among deaf or hard of hearing (DHH) adults using CI. We examined and characterised outcomes based on self-reports about treatment success and occupation status between 2018 and 2020.

### Design

We performed a prospective study using hospital clinical records and an online questionnaire with 76 DHH adults aged >15 who had received CIs since the introduction of the LRS policy in 2018. Using univariate and multivariate regression models, we investigated the relationship between demographic, audiological, and social determinants of health and outcomes, including treatment success for social inclusion (International Outcome inventory for Hearing Aids and CIs assessment: IOI-HA) and occupation status for labour market inclusion.

**Funding:** This article was supported by a full scholarship provided by the Chilean Government "Beca de Doctorado en el Extranjero Becas Chile, Convocatoria 2018, Ley N˚21.053, Asociación Nacional de Investigación y Desarrollo (ANID)". The funder had no role in study design, data collection and analysis, the decision to publish, or preparation of the manuscript.

**Competing interests:** The authors have declared that no competing interests exist.

## Results

Our study showed elevated levels of treatment success in most of the seven sub-scores of the IOI-HA assessment. Similarly, around 70% of participants maintained or improved their occupations after receiving their CI. We found a significant positive association between treatment success and market inclusion. Participants diagnosed at younger ages had better results than older participants in both outcomes. Regarding social determinants of health, findings suggested participants with high social health insurance and a shorter commute time to the clinic had better results in treatment success. For labour market inclusion, participants with high education levels and better pre- CI occupation had better post-CI occupation status.

## Conclusions

In evaluating the LRS policy for providing CIs for DHH adults in Chile, we found positive effects relating to treatment success and occupation status. Our study supports the importance of age at diagnosis and social determinants of health, which should be assessed by integrating public services and bringing them geographically closer to each beneficiary. Although evidence-based guidelines for candidate selection given by the LRS policy might contribute to good results, these guidelines could limit the policy access to people who do not meet the requirements of the guidelines due to social inequalities.

## 1. Introduction

Deafness affects population health worldwide, having devastating economic costs and health consequences [1]. The global incidence of a moderate or higher degree of hearing loss ranges from 2% for people aged 20 years to 26% among those +70 [2]. By 2050, around 1 in every four people will suffer from hearing loss with attributed economic costs of up to USD$ 2.45 billion (95% CIs 2.35–2.56) [3]. The Figs are even higher among most impoverished regions, including Latin America, where the deafness burden accounts for 4.5% of total years of healthy life lost due to disability (ranging from 3.57%-5.57%) [3]. The cost of unaddressed hearing loss for adults aged between 15–64 years was estimated at USD$ 750–790 billion in the region, according to the World Health Organization (WHO) [4]. However, the average total costs of health and education in Latin America are below the previously mentioned global benchmark, reaching USD$ 7.1 and 9.3 billion [5]. Importantly, there is wide variability within Latin-American countries, and no extensive cost analyses have been performed for this region.

DHH people can also face direct effects from their hearing loss on their social relationships with other individuals and indirect impacts on their health, psychological and economic status [6]. The latter is explained by DHH adults having greater unemployment rates, decreased incomes, lower-skilled jobs, or reduced autonomy due to reliance on family dependency [7, 8]. These features create barriers for DHH adults in the labour force, harming their communication skills and their well-being [4]. Detrimental consequences among DHH adults in Latin America are observed, compared to high-income countries (HICs). Latin American economies have one of the highest levels of informality, and effective multisectoral policies targeting improved health status among deaf people are insufficient [9]. Lack of universal access to healthcare, under-resourced hospital infrastructure, area-level deprivation, living far from the hospital, high technology costs, and unequal distribution of healthcare professionals impact

disabled people's well-being and have contributed to socioeconomic inequalities [10]. This includes people who are DHH. Without exposure to accessible sign language or appropriate rehabilitation strategies, DHH individuals' quality of life, social inclusion, and communication skills are highly compromised, especially in low-resource areas, such as in most Latin American settings [2].

A few evidence-based health policies for DHH adults have been enacted in Latin America in the last decade [2, 11]. Using cochlear implants (CI) has been a cost-effective intervention to alleviate hearing loss and consequently improve DHH people's health status, social inclusion, and labour reinsertion [12, 13]. Among adults using CI, the evaluation outcomes often consider speech perception tests and self-report evaluations of hearing and/or quality of life [14]. Nevertheless, patients' results might vary depending on their social affiliation and other social factors, such as the health-education access [15]. Still, most interventions for hearing loss in Latin America—primarily from Argentina, Brazil, Colombia, and Mexico [11]—have attempted to reach the broad population, but these only target the audiological (clinical) implications without further consideration of the social determinants of health [16, 17].

In line with the contextual and diverse factors influencing CI outcomes among the adult population, Chile introduced the "Ley Ricarte Soto" (LRS) health policy in 2018, which is a subscription package that covers 27 high-cost health conditions, including CI indicated at a post-lingual (Post-lingual: Period after spoken language acquisition.) age in adolescents and adults [18]. The policy follows international evidence-based recommendations evaluated by the Chilean Department of evidence-based healthcare and health guarantees [18, 19]. It corresponds to the first strategy employed among targeted populations to help prevent catastrophic health expenditures and reduce socioeconomic inequalities while accounting for an integrated perspective, promoting equity of access to health and social inclusion. This study evaluates the impact of the LRS policy on treatment success and labour market inclusion among Chilean DHH adults. We also examine and characterise DHH adults using CI and their outcomes based on self-reports of treatment success between 2018 and 2020.

## 2. Methods

### 2.1 Data sources and sampled population

Two Research Ethics Committees approved the study: The Faculty of Medicine, University of Chile (167–2020) and University College London (UCL) (LCD-2020-13). The approval considered data protection, procedures for collecting data and informed consent. Every participant in the study provided written informed consent. We performed a prospective study and collected data from two sources:

1. Hospital clinical records (see "S1 Appendix–Protocols" in S2 File) obtained from those adults who attended public hospitals in Chile.

2. An online survey (DHH-A Survey at Supplementary Materials in S1 File) was completed by each participant.

A bilingual committee of English and Spanish speakers created the survey, adapted it for an online format (Opinio®) and sent it via email or text to our sampled participants. Different measures of accessibility (sign language interpreters, video calls and facilitators) were put in place for participants when necessary. We invited all adults aged >15 who had been implanted since the introduction of the LRS policy between 2018 and January 2020 in public centres. Seventy-six adults were included in the study (see consort diagram "S2 Appendix- Consort diagram" in S2 File), representing 62% (76 out of 123) of all adults implanted in Chile during the

mentioned period. Exclusion criteria considered participants implanted under any other Chilean policy in public or private institutions. Table 1 summarises the variables included in our analyses with their respective description, justification, and data sources.

## 2.2 Statistical analysis

**2.2.1 Characterisation analysis.** We characterised sampled individuals according to the place where they live and their sociodemographic characteristics. First, the area where they live was plotted on a map of Chile coloured according to the BDI of each national borough (see Table 1 for more details). Second, the independent variables of the sample were described using means (and medians), standard deviations (SD) and interquartile ranges (IQR) for continuous variables. Categorical and binary variables were described by reporting their frequency and percentages within each category.

**2.2.2 Impact of the LRS policy on treatment success and labour market inclusion.** To evaluate the impact of the LRS policy on treatment success and labour market inclusion, we examined the distribution of the different variables and the bivariate associations between independent variables and our outcomes (i.e., treatment success and change in occupation status). The association between three socioeconomic variables–Education, Social Health Insurance (SHI) and SES index–and treatment success was graphically and statistically examined using boxplots and Wilcoxon t-tests. To evaluate socioeconomic factors as a variable, we standardised educational level and the SHI variables, averaging them accordingly in the SES index variable. To investigate how social determinants of health effect outcomes, we combined the seven items of the IOI-HA to produce a total score ranging from 16 to 35 [25–27]. We then performed a regression analysis. First, we fitted the data to a univariate linear regression model to explore the relationship between each independent variable and treatment and occupation outcomes. Second, variables at least presenting a significant association (p-value <0.05) or a trend toward significance (p-value between 0.05 and 0.1) were used for consecutive regression testing. We used multivariable linear regressions and logistic regressions for testing independent variables in treatment success and change in occupation outcomes, respectively. A total of four models were built for each result. Our four models were adjusted by education level, SHI, pre-occupation status and SES Index, respectively. We examined collinearity among our included variables; variables with a variance inflator factor (VIF)>5 were removed. Robust standard errors were used. All statistical analyses were performed using R Studio version 1.4.

## 3. Results

### 3.1 Descriptive analysis

Fig 1 illustrates the graphical distribution of sampled adults by Chilean boroughs using the BDI scores. The majority of our participants came from the central area (38/76, 50%) and specifically Santiago (27/76, 36%). Two participants attended rehabilitation centres situated in the northern area (3%) and 36 in the southern area (47%), primarily in Concepcion (10/76, 13%). The average BDI value varied between our sampled individuals and the country average (i.e.,0.56 and 0.37, respectively). Santiago and Concepcion have the highest BDI scores (0.78 and 0.64, respectively), whereas the remaining areas ranged below 0.60.

Table 2 shows the descriptive statistics for all sociodemographic, audiological and treatment variables for participants (n = 76). On average, participants were aged 52.23 years [SD = 17.9], and the mean age of diagnosis was 49.8 years [SD = 18.64]. Most individuals reported bilateral profound HL (88.16%). More than half (69%) of our sample belonged to "low-income" or "low-middle" levels, according to the SHI subcategories. 85% (65/76) of adults had at least completed secondary education. The most frequently reported subcategory for pre and post-

**Table 1. Dependant and independent variables.**

| Variable | Description | Justification (Effects in outcome ^) | Source |
|---|---|---|---|
| *Dependent variables* | | | |
| Treatment success | *International Outcome Inventory for Hearing Aids* (IOI-HA). Spanish-adapted version of IOI-HA [20]. Seven questions using a 5-level Likert scale by participants. Final score from 1 = Lowest score/Low success to 5 = Highest score/ quality of life. Questions regarding: 1) CI Adherence and use of the CI (Ouse), 2) Quality of life (QoL), 3) Benefits using the CI (Oben), 4) Residual activity limitations (Oral), 5) Satisfaction using the CI (Osat), 6) Residual participation restrictions (ORPR), 7) Impact on others (OioTH). | International evaluation used in adults using hearing devices. Higher scores are associated with better quality of life [20], higher use of the device [14] and improved outcomes (treatment success) (+). | DHH-A Survey |
| Occupational change | We asked participants to define their occupation status previous to and after the CI use. Three conditions levels as result; "Affected" = In case of job loss or less wage, "Maintained" = In case of same occupation/wage or in retirement process, "Improved" = In case of changing Occupation to paid job/better wage. | Improved occupational conditions are positively related to the use of CI (+) among DHH adults [18]. | DHH-A Survey |
| *Independent variables* | | | |
| Age | Continuous variable in years, starting at the age of 15 years. | Older age might have (-) effects on outcomes [21]. | Hospital clinical records |
| Diagnosis Age | Age at diagnosis, in years. | Post lingual indication of CI has (+) effect among adults using spoken language. | Hospital clinical records |
| Aetiology** | Six-level variable 1) Unknown, 2) Hereditary non-syndromic, 3) Prenatal-Perinatal, 4) Postnatal Meningitis / Infections, 5) Presbycusis, 6) Others (Trauma, Ototoxicity, Otosclerosis, Autoimmune). | Conditions limiting CI function/other abilities such as 4) and 5) has (-) effects on outcomes [14]. | Hospital clinical records |
| HL Severity | Two levels variable 1) Severe-Profound, 2) Profound. | Higher severity in HA might have (-) impact on outcomes [21]. | DHH-A Survey & Hospital clinical. Records |
| Comorbidities | Dummy variable reporting two levels; whether patients declared any chronic condition/underlying health disease*, or not. | Comorbidities has (+) impact on outcomes [14]. | Hospital clinical records |
| Borough Development index (BDI) | Index variable showing a composite number related to wellbeing and access to service across Chile [22]. It ranges from 0 (low) to 1 (high). | Index related to each territory's socioeconomic outcomes, living deprivation, and urbanisation. High index might increase outcomes (+). | Hospital clinical records |
| Social Health Insurance (SHI) level ¤ | Five level variable 1) A (very low income- poverty), 2) B (low income) 3) C (low-middle income), 4) D (middle/high income)), 5) Other (public and private Health Insurances). | SHI was assigned by health public services, according to annual incomes. Low income is related to (-) effects on CI outcomes [23, 24]. | Hospital clinical records |
| Education¤ | Three level variable: 1) Primary education completed, 2) Secondary education completed, 3) Tertiary education completed or more. | Higher education level has (+) effects on CI outcomes [23]. | Hospital clinical records |
| Previous Occupation & Post Occupation | Two level variable 1) occupation before using CI 2) occupation after using CI. The categories were 1) Employed/Student, 2) Unemployed, 3) Retired., | Pre/post-occupation has a (+) effect on communicative intent and outcome [14]. | DHH-A Survey |
| Unilateral/ Bilateral CI | Two level variable: 1) Bilateral, 2) Unilateral. | Two CI and the binaural stimulation have (+) effects [21]. | Hospital clinical records |
| Rehabilitation Attendance | Two level variable response; 1) Yes, 2) No. | Attending to rehabilitation centres might have (+) effects. | DHH-A Survey |
| Duration of Rehabilitation | Four-level variable 0) Less than 3 months, 1) 3 to 5 months, 3) 6 to 11 months, 4) 12 months or more | More time spent at rehabilitation centres might have (+) effect, due to longer treatment/rehabilitation. | DHH-A Survey |

(*Continued*)

**Table 1.** (Continued)

| Variable | Description | Justification (Effects in outcome ^) | Source |
|---|---|---|---|
| Commute time | Continuous variable in in minutes. | Living far from the rehabilitation centre might have (-) effects on the outcomes. | DHH-A Survey |

Notes: Abbreviations: HL = Hearing Loss, DHH-A Survey = Online survey to participants, Hospital C. Records = Hospital Clinical Records

(-) = Negative effect, (+) = Positive effects. Symbols: ^ = Effects in outcomes for DHH adults receiving the CI at post-lingual age, ¤ = Independent socioeconomic (SE) variables were standardised to create a socioeconomic status (SES) Index variable

* = These conditions included chronic diseases, such as hypertension, diabetes and additional conditions, such as cancer, visual impairment and others, were considered in this section.

occupation were either "workers or students", with (61.84%) and (50%), respectively. Seventy-four (97.3%) participants have a unilateral CI, while only 2 (2.63%) had bilateral devices. The majority of participants (86.64%) were attending their rehabilitation centre. Lastly, the sample spent on average 116.8 minutes commuting to their rehabilitation centre.

Overall, we observed elevated levels of success in most evaluated IOI-HA subscales. Device usage (Ouse) presented the highest score values among the subscales. Similarly, quality of life (QoL) had high results (mean = 3.98, SD = 0.93). Residual participation restrictions (Oral), derived from CI and HA, depicted the lowest score values (Median = 3, SD = 0.93). The majority of participants (70%) kept or improved their occupation status after receiving the CI.

Fig 2 displays a stacked bar graph for treatment success using IOI-HA subcategories results and labour inclusion using the occupation status results. In Fig 2A of IOI-HA subcategories, more than 75% of the participants reported scores values from 3 to 5 across all items. Quality

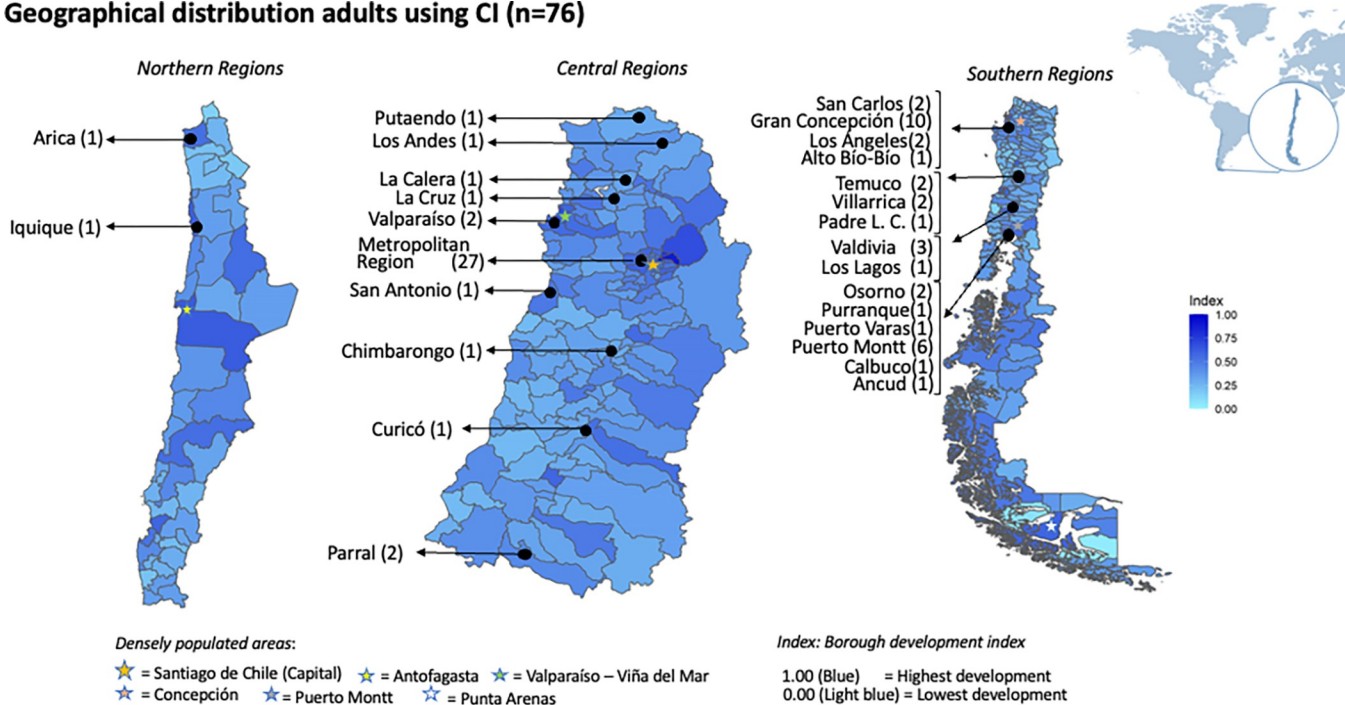

**Fig 1. Geographical distribution of adults using CI in our sample (n = 76) by BDI in Chile.** Notes: The colour scale show the Borough Develop Index [23] which evaluates the living environmental deprivation areas within the country. It merges thirteen health, social well-being, economy, and education variables in indexes from 0 to 1. Less developed boroughs are coloured in light blue, while more developed boroughs are in dark blue.

**Table 2. Descriptive statistics of dependent and independent variables (N = 76).**

| Dependent variable | | Median/Category | Mean/Freq. | [SD]/% |
|---|---|---|---|---|
| Treatment | QoL | 4 | 3.98 | [0.93] |
| Success | Ouse | 5 | 4.71 | [0.68] |
| | Osat | 5 | 4.36 | [0.86] |
| IOI-HA | ORPR | 4 | 3.52 | [1.20] |
| | Oral | 3 | 3.07 | [0.93] |
| | OioTH | 4 | 3.40 | [1.20] |
| | Oben | 4 | 3.77 | [1.15] |
| Labour market inclusion | | Improved | 28 | 37.33 |
| Occupation status | | Maintained | 31 | 41.34 |
| | | Affected | 16 | 21.33 |
| **Independent Variable** | | **Category** | **Mean/Freq.** | **[SD]/%** |
| Age | | From 15 up to 82 years | Mean: 52.23 | [17.90] |
| Diagnosis age | | From 1 up to 65 years | Mean: 49.8 | [18.64] |
| Aetiology | | Late hearing loss | 36 | 47.37 |
| | | Hereditary non-syndromic | 9 | 11.84 |
| | | Prenatal-Perinatal | 1 | 1.32 |
| | | Postnatal: Meningitis/Infections | 7 | 9.21 |
| | | Postnatal: Presbyacusis | 1 | 1.32 |
| | | Postnatal: Others | 22 | 28.95 |
| HL Severity | | Severe-to-profound HL | 9 | 11.84 |
| | | Profound HL | 67 | 88.16 |
| Underlying health conditions * | | None declared | 35 | 46.05 |
| | | Registered/declared | 41 | 53.95 |
| BDI | | Index from 0 up to 1 | Mean:0.51 | [0.11] |
| SHI | | Low income | 7 | 9.59 |
| | | Low-middle income | 44 | 59.67 |
| | | Middle income | 9 | 12.33 |
| | | Middle—high Income | 16 | 19.18 |
| Education | | Primary | 11 | 14.47 |
| | | Secondary | 51 | 67.11 |
| | | Tertiary | 14 | 18.42 |
| SES Index | | From -1.48 up to 1.67 | Mean: 0.0 | [0.83] |
| Previous Occupation | | Employed/Student | 47 | 61.84 |
| | | Unemployed | 29 | 38.16 |
| | | Retired | 0 | 0 |
| Post Occupation | | Employed/Student | 38 | 50.67 |
| | | Unemployed | 27 | 36 |
| | | Retired | 10 | 13.33 |
| Uni/Bilateral CI | | Bilateral | 2 | 2.63 |
| | | Unilateral | 74 | 97.3 |
| Rehabilitation attendance | | No | 10 | 13.15 |
| | | Yes | 66 | 86.84 |
| Rehabilitation time | | Less than 30 m | 12 | 18.18 |
| | | 30 to 50 m | 7 | 10.61 |
| | | 60 to 110 m | 14 | 21.21 |
| | | 120m and more | 33 | 50 |

*(Continued)*

**Table 2.** (Continued)

| Commute time in minutes | From min 10 max 210 | 116.80 | [77.43] |
|---|---|---|---|

Notes: IOI-HA abbreviations (Score Min = 1, Max = 5): 1) Qol = Quality of life (QoL), 2) OUse = CI Adherence and use the CI, 3) OSat = Satisfaction using the CI, 4) ORPR = Residual participation restrictions, 5) Oral = Residual activity limitations, 6) OloTH = Impact on others, 7) OBen = Benefits using the CI. BDI = Borough Development index, Rehab. Attend = Rehabilitation attendance. SD = Standard deviation. Freq. = Frequency

\* These conditions included chronic diseases, such as hypertension, diabetes, and additional conditions, such as cancer, visual impairment and others were considered in this section.

of life (QoL), device use (Ouse) and satisfaction with the device (OSat) had high results, with least than 10% of the lowest scores in all the cases. On the contrary, specific limitations given by the severity of HA, such as participation restriction (ORal) and impact on communication with others (OloTH) showed poor results in our sample.

In Fig 2B, most participants maintained their occupation status (41%), while 37% improved their occupation. Only 21% (16/76) of the participants perceived that their occupation status was affected.

Fig 3 illustrates the relation of treatment success scores with three socioeconomic variables: Education Level, SHI and SES index in tertiles. We observe an increased score among upper socioeconomic levels in all analysed variables, regardless of the referred socioeconomic variable. Fig 3A shows a trend toward a difference between primary (reference category) and tertiary education (Wilcoxon test p-value = 0.086). In Fig 3B, there a significant difference between low-income (reference category) and middle-high levels of SHI and a trend towards a difference between low-income and low-middle SHI. Although there was a higher median score among SHI index tertiles, we did not find a significant difference.

## 3.2 Statistical analysis

Table 3 displays the results of the univariate regression analysis. We found a direct association between treatment success and "diagnosis Age", "middle-high income" level of SHI, "previous occupation", and "commuting time" variables. There was a trend towards an association between those individuals having tertiary education and treatment success. Only previous occupation status showed an association with change in occupation (p = 0.003).

Table 4 depicts the results of the multivariable regressions, displaying four—outcome specific—models. We did not find an association between education and treatment success (Model 1, β = 1.26, 95%CI = -1.36, 3.89, p-value = 0.341). In Model 2, we found a trend towards an association between SHI's middle-high levels and treatment success (β = 3.74, 95% CI = -0.31–7.78, p-value = 0.070). In Models 3 and 4, we did not find any significant association with the socioeconomic variables measured. There was a trend towards a negative association between the age of diagnosis and treatment success in Model 2 (β = 0.05, 95%CI = -0.10–0.01, p-value = -0.078) and Model 4 (β = 0.01, 95%CI = -0.02–0.00, p-value = 0.093). Commuting time to the rehabilitation centre also showed a trend towards a negatively association with treatment success in model 1 and model 4, (β = -0.01, 95%CI = -0.03–0.00, p-value = 0.086; β = -0.01, 95%CI = - -0.02–0.00, p-value = 0.093, respectively). For change in occupation status, we found a significant association between "age of diagnosis" and change in occupation in all proposed models; Model 1 (OR = 0.034, SE = 0.014, p–value = 0.015), model 2 (OR = 0.030, SE = 0.013, p–value = 0.017), model 3 (OR = 0.003, SE = 0.013, p–value = 0.011) and model 4 (OR = 0.028, SE = 0.013, p–value = 0.028). We also found a significant association between preoccupation status and occupation status in model 3 (OR = 1.139, SE = 0.495, p–

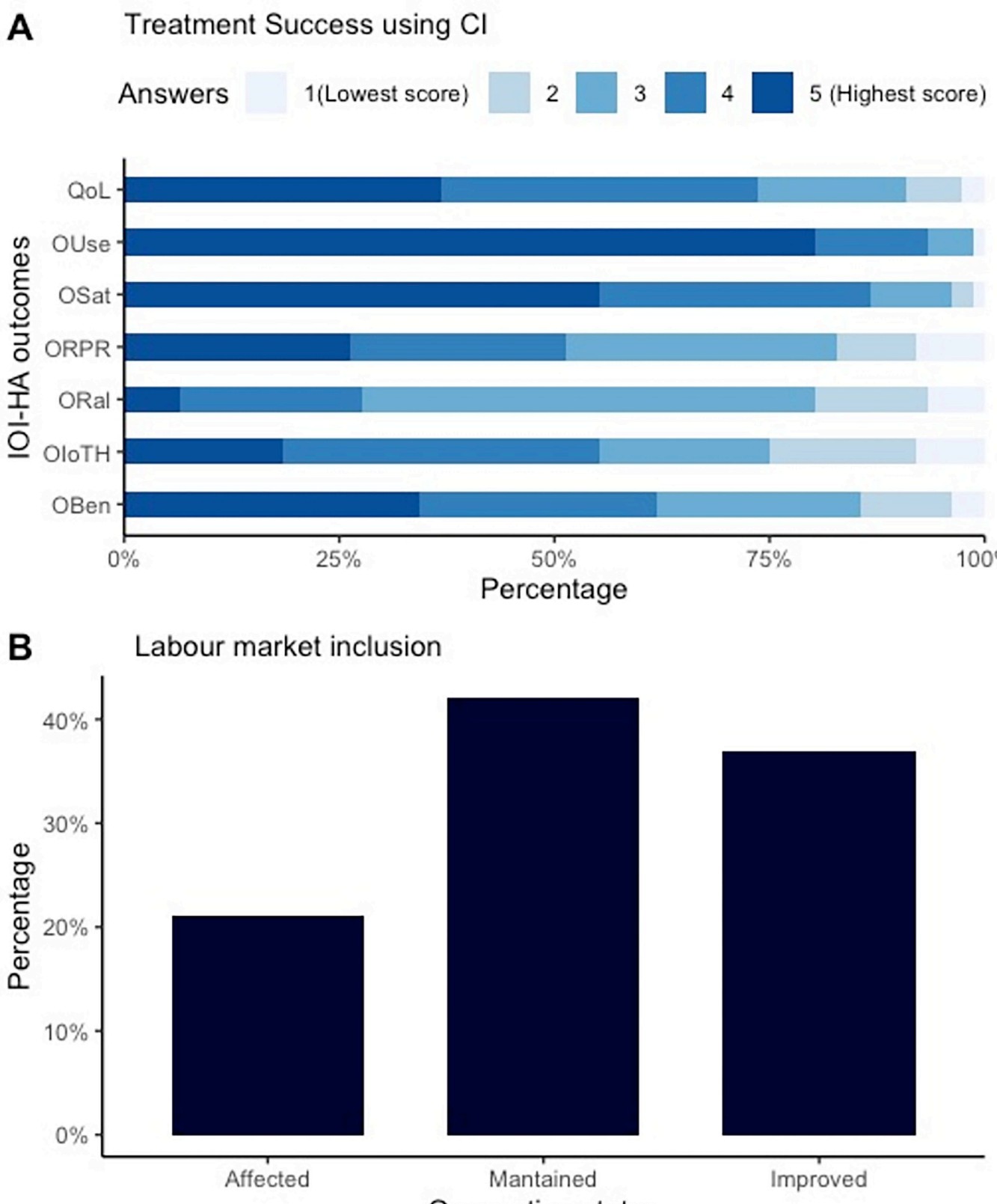

**Fig 2. Treatment success and change in occupation status among adults using CI (N = 76).** Notes: CI = Cochlear implant. IOI-HA abbreviations (Score Min = 1, Max = 5): 1) Qol = Quality of life (QoL), 2) OUse = CI Adherence and use the CI, 3) OSat = Satisfaction using the CI, 4) ORPR = Residual participation restrictions, 5) Oral = Residual activity limitations, referring to restricted activities given by the HL, 6) OloTH = Impact on others, 7) OBen = Benefits using the CI.

value = 0.021).A trend towards a positive association was found between the change in occupation status and having secondary education (OR = 1.079, SE = 0.594, p–value = 0.069).

## 4. Discussion

This study evaluated the impact of a Chilean high-cost health policy on treatment success and occupational change in the first two years of implementation among a representative number of beneficiaries (62%). High-cost health policy evaluations are particularly important for Chile and other emerging countries, which need to use limited resources for better and more efficient health policies [28]. The evaluation yielded positive impacts of this policy upon social and labour inclusion.

Our results align with previous evidence [14, 21, 25, 29–31] showing positive results in DHH adults with postlingual CI treatment. The outcome of treatment success measure (IOI-HA) reported high median values for most of the subtest scores. For example, in the current sample, "CI use" median value was 5 (the maximum score), equivalent to a constant and consistent use of the device. Device usage rates in our study align with those previously observed in different studies showing high use levels in adult users [32]. Better communication outcomes are correlated with higher device use [33]. On the other hand, some of the 'residual activity limitations (oral)' subtest scores showed a low value for the median score, which is driven by higher levels of hearing loss among users with CI. A negative correlation between the degree of HL and residual activity limitations in users' daily life has also been reported elsewhere [34].

Concerning occupational change, although it was not possible to find previous evidence measuring this outcome, previous studies also have suggested a positive impact of CI on

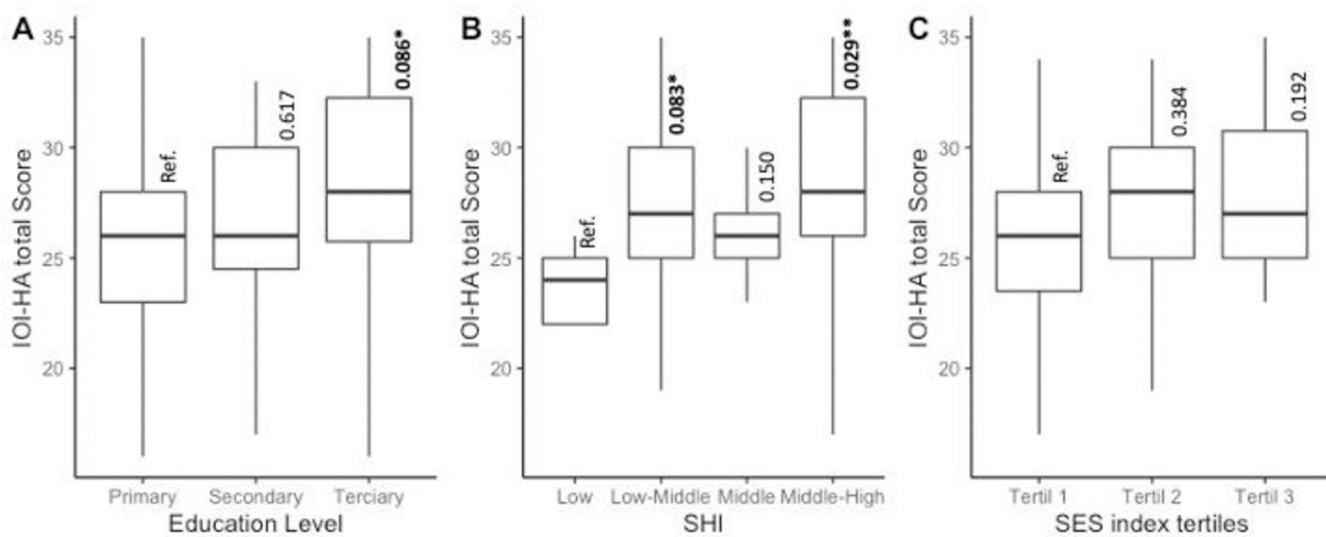

**Fig 3. Treatment success rates, by socioeconomic background.** Notes: A = Box plot between Treatment Success (IOI-HA score) and three levels of education. B = Box plot between Treatment Success and four levels of SHI. Fig 2C = Box plot between Treatment Success (IOI-HA score) outcome score and four levels of SHI. C = Box plot between Treatment Success outcome score and SES INDEX tertiles. Abb: Alpha = p-value 0.05 '**' 0.1 '*', Ref. = Reference category. SHI = Social Health Insurance, SES = Socioeconomic levels. Wilcoxon tests were employed to explore differences across each variable's subcategories.

**Table 3. Univariate linear regression results (N = 76).**

| Predictors | Treatment success (score) | | | Change in occupation status | | |
|---|---|---|---|---|---|---|
| | β | 95% CI | p-value | Estimates | 95% CI | p-value |
| *Demographic variables* | | | | | | |
| Age | -0.03 | -0.09 – 0.02 | 0.241 | -0.00 | -0.01 – 0.00 | 0.197 |
| Sex (Male) | 1.49 | -0.62 – 3.60 | 0.164 | 0.04 | -0.15 – 0.23 | 0.679 |
| Age at diagnosis | -0.06 | -0.11 – -0.01 | **0.031**\*\* | -0.13 | -0.31 – 0.06 | 0.186 |
| Other health conditions | -0.00 | -2.12 – 2.12 | 0.997 | 0.33 | -0.30 – 0.97 | 0.301 |
| Aetiology | | | | | | |
| Meningitis | 3.37 | -1.28 – 8.01 | 0.153 | -0.10 | -0.51 – 0.32 | 0.646 |
| Late HA | 1.36 | -2.07 – 4.79 | 0.432 | 0.19 | -0.11 – 0.50 | 0.206 |
| Presbycusis | 2.22 | -4.98 – 9.42 | 0.540 | 0.33 | -0.30 – 0.97 | 0.301 |
| Others | 0.22 | -3.42 – 3.87 | 0.904 | 0.11 | -0.22 – 0.43 | 0.514 |
| *Socioeconomic variables* | | | | | | |
| Education level (Ref.: primary) | | | | | | |
| Secondary | 0.51 | -2.02 – 3.04 | 0.687 | 0.10 | -0.13 – 0.33 | 0.379 |
| Tertiary | 2.33 | -0.17 – 4.83 | **0.068**\* | 0.11 | -0.12 – 0.34 | 0.341 |
| SHI (Ref.: low income) | | | | | | |
| SHI Low-middle income | 2.86 | -0.80 – 6.53 | 0.123 | -0.13 | -0.46 – 0.20 | 0.441 |
| SHI Middle income | 1.89 | -2.65 – 6.42 | 0.409 | 0.03 | -0.38 – 0.45 | 0.879 |
| SHI Middle-high income | 4.63 | 0.55 – 8.70 | **0.027**\*\* | 0.02 | -0.35 – 0.39 | 0.924 |
| SES index | 1.54 | 0.30 – 2.77 | **0.015** \*\* | 0.07 | -0.04 – 0.19 | 0.211 |
| Previous Occupation | -2.16 | -4.28 – -0.05 | **0.045**\*\* | 0.28 | 0.10 – 0.47 | **0.003**\*\* |
| Commute Time Hrs | -0.01 | -0.03 – -0.00 | **0.038**\*\* | 0.00 | -0.00 – 0.00 | 0.256 |
| Borough Development Index | -2.65 | -12.81 – 7.50 | 0.604 | 0.14 | -0.77 – 1.05 | 0.755 |
| Rehabilitation status | 2.48 | -0.59 – 5.56 | 0.112 | -0.13 | -0.41 – 0.15 | 0.364 |
| Rehab time (Ref.:15–30 min) | | | | | | |
| Rehabilitation 30–50 min | -1.02 | -4.24–2.20 | 0.529 | -0.00 | -0.30 – 0.29 | 0.980 |
| Rehabilitation 60–110 min | 1.08 | -1.55 – 3.72 | 0.416 | -0.03 | -0.27 – 0.21 | 0.791 |
| Rehabilitation >120 min | -1.24 | -4.81 – 2.34 | 0.492 | 0.11 | -0.21 – 0.43 | 0.499 |

Notes: Late HA: Late Hearing loss, SHI: Social Health Insurance, SES index: SES standardisation explained in Table 2. Alpha = p-value 0.05

'\*\*' 0.1

'\*'. Rehab Time: Rehabilitation time per session.

employment status and labour conditions [31], indicating higher levels of work satisfaction after having a CI. In our study, many participants considered themselves "unemployed" when doing unpaid care work at home. This is a regular occupation in countries such as Latin America, but it is not considered a formal job. Further research using labour inclusion outcomes needs to be complemented with more options for unpaid caring work. Future studies could use measures of salary range or well-being at work [35].

Our study suggests the importance of early HL diagnosis for better results in treatment success and occupation status. Previous evidence suggests that adults with long-standing DHH without treatment might receive fewer benefits than those diagnosed and aided rapidly [36]. Although elderly patients can receive the same benefits as younger individuals in using CI [37], a longer time between diagnosis and implantation can lead to higher risks of social exclusion and ageing effects [38]. Although positive outcomes have been reported in adults who attend ongoing rehabilitation [23], in the current study, most participants attended rehabilitation at most only during their first year following cochlear implantation. In this study, we did

**Table 4. Multivariate regression analyses for the association between our outcomes and sociodemographic variables (n = 76).**

**Outcome 1 = Treatment success**

| Predictors | Model 1: + Education | | | Model 2: + SHI | | | Model 3: + Pre-Occupation | | | Model 4: + SES INDEX | | |
|---|---|---|---|---|---|---|---|---|---|---|---|---|
| | Est. | 95% CI | p-value | Est. | 95% CI | p-value | Est. | 95% CI | p-value | Est. | 95% CI | p-value |
| Age at diagnostic | -0.05 | -0.10 , 0.01 | 0.124 | -0.05 | -0.10 , 0.01 | **0.078***  | -0.05 | -0.10 , 0.01 | **0.079***  | -0.04 | -0.09 – 0.02 | 0.186 |
| Commute time | -0.01 | -0.03 , 0.00 | **0.086***  | -0.01 | -0.02 , 0.00 | 0.146 | -0.01 | -0.02 , 0.00 | 0.109 | -0.01 | -0.02 – 0.00 | **0.093***  |
| Education level (Ref. = primary) | | | | | | | | | | | | |
| –Secondary | -0.23 | -2.79 , 2.34 | 0.862 | | | | | | | | | |
| Tertiary | 1.26 | -1.36 , 3.89 | 0.341 | | | | | | | | | |
| SHI (Ref. = Low) | | | | | | | | | | | | |
| Low middle | | | | 2.52 | -1.11 , 6.15 | 0.171 | | | | | | |
| Middle | | | | 1.61 | -2.82 , 6.03 | 0.471 | | | | | | |
| Middle high | | | | 3.74 | -0.31 , 7.78 | **0.070***  | | | | | | |
| Preoccupation status (Ref. = E/S) | | | | | | | -1.64 | -3.73 , 0.46 | 0.124 | | | |
| SES index | | | | | | | | | | 1.11 | -0.17 – 2.39 | **0.089***  |
| Constant | 29.21 | 26.07, 32.36 | **<0.001** | 27.02 | 22.88, 31.16 | **<0.001** | 31.78 | 28.42 , 35.15 | **<0.001** | 29.26 | 26.91 – 31.61 | **<0.001** |

**Outcome 2 = Labour market inclusion Δ**

| Predictors | Model 1: + Education | | | Model 2: + SHI | | | Model 3: + Pre-Occupation | | | Model 4: + SES INDEX | | |
|---|---|---|---|---|---|---|---|---|---|---|---|---|
| | OR | SE | p-value | OR | SE | p-value | OR | SE | p-value | OR | SE | p-value |
| Age at diagnostic | 0.034 | 0.014 | **0.015**** | 0.030 | 0.013 | **0.017** | 0.003 | 0.013 | **0.011**** | 0.028 | 0.013 | **0.028**** |
| Commute time | 0.001 | 0.003 | 0.746 | -0.001 | 0.003 | 0.646 | 0.0 | 0.003 | 0.954 | 0.001 | 0.003 | 0.693 |
| Education level (Ref. = primary) | | | | | | | | | | | | |
| Secondary | 1.079 | 0.594 | **0.069***  | | | | | | | | | |
| Tertiary | 0.049 | 0.574 | 0.932 | | | | | | | | | |
| SHI (Ref. = Low) | | | | | | | | | | | | |
| Low middle | | | | 0.135 | 0.806 | 0.867 | | | | | | |
| Middle | | | | 0.547 | 1.007 | 0.587 | | | | | | |
| Middle-high | | | | 0.078 | 0.897 | 0.860 | | | | | | |
| Pre-occupation status (Ref. = E/S) | | | | | | | 1.139 | 0.495 | **0.021**** | | | |
| SES index | | | | | | | | | | 0.008 | 0.287 | 0.978 |
| Constant D/I | 0.625 | 0.736 | 0.396 | 0.159 | 0.900 | 0.860 | 1.165 | 0.753 | 0.122 | 0.130 | 0.526 | 0.805 |
| Constant D/M | 1.578 | 0.752 | **0.036** | 1.080 | 0.908 | 0.234 | 0.194 | 0.741 | 0.794 | 1.043 | 0.539 | **0.053** |

Notes: Δ: Ordinal regression. Abb: Dg Age = Diagnosis Age, Com T. = Commute Time, Ed–Sec = Secondary Education, Ed–Ter = Tertiary Education, SHI LM = Low-Middle SHI, SHI M = Middle SHI, SHI MH = Middle High SHI, Ref = Reference, Pre-Occ. = Previous Occupation, E/S = Employed/Student, SES INDEX = SES standardisation, Constant = Regression intercept, D/I = Improves/ Diminished occupation status, D/M = Diminished/Maintains. **Alpha** = p-value **0.05**

'**'* 0.1

'*' OR = Odds rat

not assess the impact of aetiologies because most of our sample had late-onset hearing loss. Future studies need to consider a diversity of aetiologies.

This is, to our knowledge, the first study showing a Chile-wide picture regarding the impact of CI upon specific social determinants of health and outcomes expected for DHH adults using CI. The results suggest that high SHI positively affects treatment success. Workers receiving greater salaries had advantageous health insurance and improved healthcare access [24]. The association between better-paid jobs due to better work qualifications can account for the influence of better health insurance on treatment success. This reinforces the importance of increasing the opportunities for labour inclusion and improving DHH adults' occupation and work skills. Even in developed countries like the US, only 53.3% of deaf adults are

**Table 5. Table of costs for cochlear implant treatments among adult patients in Chile.**

| Item | 1st year | 2nd and 5th year (processor change) |
|---|---|---|
| Cochlear Implant Standard Kit | USD$12550 (CLP$ 9389000) | NC |
| Surgery | USD$2880 (CLP$ 2160000) | NC |
| Hospital expenses in surgery | USD$3720 (CLP$ 2791000) | NC |
| Medical exams | USD$1270 (CLP$ 950000) | $USD110 (CLP$ 80000) |
| Medical appointments after surgery | USD$300 (CLP$ 220000) | $USD160 (CLP$120000) |
| Audiological appointments | USD$160 (CLP$ 120000) | $USD110 (CLP$80000) |
| Calibration and training sessions | USD$1070 (CLP$ 800000) | USD$110 (CL$80000) |
| Accessories replacement (Processor after five years) | NC | USD$7700 (CLP$5770000) |
| Transport to the health centre | USD$200 (CLP$ 150000) | USD$80 (CLP$100000) |
| Other costs (Legal documents, personal care supplies, etc.) | USD$200 (CLP$150000) | USD$40 (CL$50000) |
| **Total, by year** | USD$22350 (CLP $16730000) | USD$8200 (CLP$ 6280000) |
| **Total** | USD$30550 (CLP$23010000) | |

Notes: The costs are presented in approximated USD$ and (CLP$), rate exchange USD$1 = CLP$750 (April 2022). Abb: NC = Not available. Source: Our research in hospitals from the Metropolitan Region of Chile and Public National Medical Store CENABAST www.cenabast.cl.

employed compared with 75.8% of hearing people [39]. This gap in the employment rate between deaf adults and hearing people is more significant in countries with emerging economies or where there are vast inequalities, possibly reflecting the lack of training opportunities and requirements for higher work skills [4].

Although better education has been reported as beneficial for better CI outcomes, our study did not find a strong relationship between education and either treatment success or occupation status. The CI candidate selection process in the LRS high-cost policy can explain this lack of association, as it includes specific social, educational, and psychological requirements [18]. More than 80% of our sample have at least secondary education completed, which is exceptionally high considering the average in the country [40]. Similarly, the higher BDI mean in our sample (mean = 0.51, SD = 0.11) compared with the national average (0.34) demonstrates that our participants had more advantageous conditions. This represents a challenge to the LRS policy. The requirements for inclusion as a CI candidate in the Chilean LRS policy are likely to exclude many candidates in a country with critical social inequalities [10]. This is in line with findings from previous studies in Chile with adults using hearing aids, suggesting an association between higher education and socioeconomic variables and better social support and positive attitudes towards hearing loss or hearing aids [41].

Our study found a relationship between shorter commute times and favourable outcomes for treatment success and occupation status. Although our outcomes do not directly relate to living environment deprivation areas in Chile, the possible importance of the commute time suggests a challenge shared in Chile and Latin America. International recommendations in health indicate the importance of transferring the rehabilitation process and health control to the primary health centres with shorter commutes [42]. Previous studies in rehabilitation have shown better results following treatment when there is less commute time to the health centre [42, 43]. Considering this challenge, the Chilean Ministry of Health launched a national plan

for 2021–2030 for hearing health and ear care, focusing on creating new policies and programs for better and closer health/rehabilitation for hearing loss in children and adults [44]. Emphasising the role of the primary health sector aligns with the World Health Organization's recommendations to tackle the impact of SES on people from countries worldwide [2]. Additionally, collaboratively working among stakeholders, policymakers, researchers and clinicians on policies for better access and outcomes in patients is essential [1].

As an exploratory analysis, we collected data on the treatment and rehabilitation cost of the CI intervention based on expert knowledge and the Chilean public national central medical store (www.cenabast.cl) (Table 5). Our total cost (US$30,550) is in line with other high-income countries, where total costs stand at US$35,000 in France [45] and to US$52,000 in Switzerland [46]. Although the CI has been evaluated as cost-effective in some middle-income countries [12], the policy benefits in Chile need to be assessed in detail. Inequalities, lack of funding and social protection programs might influence cost-effectiveness [45]. However, benefits using quality-adjusted life years (QALYs) in countries such as the Netherlands have shown significant health benefits of the CI, rising up to €275,000 (95% UI = –€110,000; €604,000) per user [12]. Total costs per user were USD$30550, and we observed more than 75% of CI beneficiaries with high or very high ranking for quality of life in subtest scores of treatment success, which indicates significant health and cost benefits.

This study has some shortcomings. First, although we had a representative number of participants (76 out of a maximum 123), the sample size could potentially affect our statistical analyses. Second, it is crucial to consider expected bias due to the self-report survey in participants. Third, more detailed data about living environment deprivation areas per participant would be necessary to assess its relationship with the expected outcomes fully. The study does have some considerable strengths. It is the first evaluation of a national sample of DHH adults using CI in a Latin-American country, characterising more than 60% of the national population implanted during the first two years of the new policy. Similarly, this is the first assessment of a high-cost policy implemented in Chile under the LRS policies, which considers 27 expensive health conditions [18] from 2018. Our results have implications for other emerging countries with high-cost policies providing cochlear implants, where health and social impact is crucial for ensuring better results.

## 5. Conclusion

This novel evaluation of the high-cost LRS policy for DHH adults receiving CI in Chile showed positive results in line with the policy aims of improving social and labour market inclusion. Beneficiary selection requirements could explain the positive results. Although the findings are successful from the policy analysis perspective, there is a challenge for those potential beneficiaries who are not achieving the policy requirements due to inequality conditions. Our findings are in line with previous evidence supporting the importance of age of diagnosis and social determinants of health in delivering positive outcomes. In addition, our findings suggest that consideration should be given to the positive impact of integrated health services, which could shorten commute time and reduce inequality of access to public services in highly economically unequal countries, such as Chile. The above supports the importance of policies based on the primary care service and the social determinants of health, with programs produced collaboratively among stakeholders, policymakers, researchers, and clinicians.

## Supporting information

**S1 File.**
(DOCX)

**S2 File.**
(DOCX)

## Acknowledgments

All authors attest they meet the ICMJE criteria for authorship and have reviewed and approved the final article. This article was supported by a full scholarship provided by the Chilean Government "Beca de Doctorado en el Extranjero Becas Chile, Convocatoria 2018, Ley N˚21.053, Asociación Nacional de Investigación y Desarrollo (ANID)". The funder had no role in study design, data collection and analysis, the decision to publish, or preparation of the manuscript.

## Author Contributions

**Conceptualization:** Mario Bustos-Rubilar, Fiona Kyle, Eliazar Luna, Daniel Tapia-Mora, Merle Mahon.

**Data curation:** Mario Bustos-Rubilar, Fiona Kyle, Eliazar Luna, Kasim Allel, Ximena Hormazabal, Daniel Tapia-Mora, Merle Mahon.

**Formal analysis:** Mario Bustos-Rubilar, Kasim Allel, Ximena Hormazabal.

**Investigation:** Mario Bustos-Rubilar, Fiona Kyle, Kasim Allel, Daniel Tapia-Mora, Merle Mahon.

**Methodology:** Mario Bustos-Rubilar, Fiona Kyle, Eliazar Luna, Kasim Allel, Ximena Hormazabal, Daniel Tapia-Mora, Merle Mahon.

**Project administration:** Mario Bustos-Rubilar, Kasim Allel, Daniel Tapia-Mora.

**Resources:** Mario Bustos-Rubilar, Eliazar Luna.

**Software:** Mario Bustos-Rubilar, Eliazar Luna.

**Supervision:** Mario Bustos-Rubilar, Fiona Kyle, Merle Mahon.

**Visualization:** Mario Bustos-Rubilar, Eliazar Luna, Kasim Allel.

**Writing – original draft:** Mario Bustos-Rubilar, Eliazar Luna, Kasim Allel, Ximena Hormazabal, Daniel Tapia-Mora.

**Writing – review & editing:** Mario Bustos-Rubilar, Fiona Kyle, Eliazar Luna, Kasim Allel, Daniel Tapia-Mora, Merle Mahon.

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
