## [Decision Letter · Decision Letter 0]

19 May 2023

A country-wide health policy in Chile for deaf adults using cochlear implants: analysis of health determinants and social impacts

PONE-D-23-04663

Dear Dr. %Bustos Rubilar%,

We’re pleased to inform you that your manuscript has been judged scientifically suitable for publication and will be formally accepted for publication once it meets all outstanding technical requirements.

Kind regards,

Mary Diane Clark, PhD

Academic Editor

PLOS ONE

Journal Requirements:

"This article was supported by a full scholarship provided by the Chilean Government "Beca de Doctorado en el Extranjero Becas Chile, Convocatoria 2018, Ley N°21.053, Asociación Nacional de Investigación y Desarrollo (ANID)" to MB.  

" ext-link-type="uri" xlink:type="simple">https://www.anid.cl/capital-humano/doctorado-en-el-extranjero-becas-chile/"

Please respond by return e-mail so that we can amend your financial disclosure and competing interests on your behalf.

5. We note that Figure 1 in your submission contain map images which may be copyrighted. All PLOS content is published under the Creative Commons Attribution License (CC BY 4.0), which means that the manuscript, images, and Supporting Information files will be freely available online, and any third party is permitted to access, download, copy, distribute, and use these materials in any way, even commercially, with proper attribution. For these reasons, we cannot publish previously copyrighted maps or satellite images created using proprietary data, such as Google software (Google Maps, Street View, and Earth). For more information, see our copyright guidelines: http://journals.plos.org/plosone/s/licenses-and-copyright.

(1) You may seek permission from the original copyright holder of Figure 1 to publish the content specifically under the CC BY 4.0 license.  

6. Please include a complete copy of PLOS’ questionnaire on inclusivity in global research in your revised manuscript. Our policy for research in this area aims to improve transparency in the reporting of research performed outside of researchers’ own country or community. The policy applies to researchers who have travelled to a different country to conduct research, research with Indigenous populations or their lands, and research on cultural artefacts. The questionnaire can also be requested at the journal’s discretion for any other submissions, even if these conditions are not met. Please find more information on the policy and a link to download a blank copy of the questionnaire here: https://journals.plos.org/plosone/s/best-practices-in-research-reporting. Please upload a completed version of your questionnaire as Supporting Information when you resubmit your manuscript.

**Additional Editor Comments:**

Two reviewers have provided feedback to accept your paper as written. I support their decision and am recommending acceptance to the journal. Sorry that this took a while. I struggled to get reviewers.

Reviewers' comments:

Reviewer's Responses to Questions

**Comments to the Author**

1. Is the manuscript technically sound, and do the data support the conclusions?

Reviewer #1: Yes

Reviewer #2: Yes

2. Has the statistical analysis been performed appropriately and rigorously? 

Reviewer #1: Yes

Reviewer #2: Yes

3. Have the authors made all data underlying the findings in their manuscript fully available?

Reviewer #1: Yes

Reviewer #2: Yes

4. Is the manuscript presented in an intelligible fashion and written in standard English?

Reviewer #1: Yes

Reviewer #2: Yes

5. Review Comments to the Author

Reviewer #1: It seems to me a very well thought out and developed study. It is clearly written and exposes the results of a relatively recent public health measure in Chile. Knowing the good results of this law supports the financing of hearing rehabilitation in Chile with the possibility of extrapolating this option to other countries in the region. On the other hand, the good results speak in favor of the inclusion criteria to access the benefit.

Reviewer #2: The narrative in clearly written and provides a succinct description of the impact of cochlear implants on the socialization skills for Chilean CI users. Also discussed are the disparities in access to CI. Statistical analyses are appropriate as applied. Data and findings are presented in multiple formats, allowing the reader to view results of the study in different forms (narrative and visual). The narrative is easy to read and presents information in a manner which is easy to understand.

6. PLOS authors have the option to publish the peer review history of their article (what does this mean?). If published, this will include your full peer review and any attached files.

Reviewer #1: **Yes: **Carolina Der

Reviewer #2: No

---

## [Editor Report · Acceptance letter]

4 Jul 2023

PONE-D-23-04663 

A country-wide health policy in Chile for deaf adults using cochlear implants: analysis of health determinants and social impacts 

Dear Dr. Bustos-Rubilar:

I'm pleased to inform you that your manuscript has been deemed suitable for publication in PLOS ONE. Congratulations! Your manuscript is now with our production department. 

Kind regards, 

on behalf of

Dr. Mary Diane Clark 

Academic Editor

PLOS ONE